# A Review of the Pro-Health Activity of *Asparagus officinalis* L. and Its Components

**DOI:** 10.3390/foods13020288

**Published:** 2024-01-16

**Authors:** Beata Olas

**Affiliations:** Department of General Biochemistry, Faculty of Biology and Environmental Protection, University of Lodz, Pomorska 141/3, 90-236 Lodz, Poland; beata.olas@biol.uni.lodz.pl; Tel./Fax: +48-42-6354485

**Keywords:** asparagus, *Asparagus officinalis* L., healthy

## Abstract

The genus *Asparagus* comprises about 300 species, including *A. curilus*, *A. filicinus*, *A. reacemosus*, and *A. officinalis* L. A particularly well-known member of the genus is *Asparagus officinalis* L., also known as “the king of vegetables”. Consuming *A. officinalis* makes an excellent contribution to a healthy diet. Modern studies have shown it to have a diuretic effect and promote defecation; it also demonstrates high levels of basic nutrients, including vitamins, amino acids and mineral salts. Moreover, it is rich in fiber. Asparagus contains large amounts of folic acid (10 cooked shoots provide 225 micrograms, or almost 50% of the daily requirement) and vitamin C (10 cooked shoots provide 25 mg). The present review describes the current literature concerning the pro-health properties of various parts of *A. officinalis* L., with a particular focus on its spears. It is based on studies identified in electronic databases, including PubMed, ScienceDirect, Web of Knowledge, Sci Finder, Web of Science, and SCOPUS. The data indicate that the various parts of *A. officinalis*, especially the spears, contain many bioactive compounds. However, although the extracts and chemical compounds isolated from *A. officinalis*, especially saponins, appear to have various biological properties and pro-health potential, these observations are limited to in vitro and animal in vivo models.

## 1. Introduction

The genus *Asparagus*, the name being derived from the Greek word meaning *stalk* or *shoot*, comprises about 300 species, including *A. curilus*, *A. filicinus*, and *A. reacemosus*. The roots are valued by herbalists for their biological activity; for example, *A. filicinus* is used to treat rheumatism. Recently, Majumdar et al. [1] reported that *A. racemosus* demonstrates neuro-nutraceutical potential, and that the extracts restore neurotransmitter activity and prevent oxidative damage in neurons.

A particularly well-known member of the genus is *Asparagus officinalis* L., also known as “the king of vegetables”. This perennial species occurs naturally around the Mediterranean area. It is the only edible representative of the genus *Asparagus* and has been cultivated all over the world. In 2018, total production covered 57,000 ha in China, mainly green asparagus, and around 65,000 ha in Europe, including both green and white asparagus. The largest importers of asparagus are the USA, importing over 92,000 tons, Germany, 40,000 tons, and Japan, 17,000 tons.

*A. officinalis* is valued for its taste and ease of digestibility, particularly the edible young shoots, commonly called *asparagus* or *asparagus sprouts*. Both white and green sprouts can be found; however, these are from the same species: green protrusions grow above the surface in the presence of light, and white ones grow covered with earth, i.e., in the absence of light. The spears and stems are usually processed into canned products in the food industry.

Asparagus is characterized by a delicate taste and is often described as one of the tastiest vegetables. They are used in salads and soups, but they also serve as an addition to meat and fish dishes, poultry and other vegetables. In addition, their roasted seeds are a replacement for coffee. Interestingly, some countries, including Peru, Belgium and the Netherlands, prefer white asparagus, whereas other countries, such as the UK, favor the green form.

The popularity of asparagus is arguably due to its flavor. White asparagus (i.e., “white gold”) is usually considered to have a milder and more delicate taste than the green type. The flavor is often reported as being similar to grassy notes in wines, including Sauvignon Blanc. However, its final flavor is a rather complex combination of aroma and taste [2]. The key odorants in the flavor of cooked asparagus are given in Table 1.

It was cultivated as a vegetable by the ancient Egyptians, Greeks and Romans. Its images are present on Egyptian sarcophagi from 5000 years ago. Indeed, Cato the Elder (d. 149 BC) recorded asparagus in *De agri cultura* as the only vegetable next to cabbage worth growing. Columella, in the first century AD, wrote that the Romans preserved asparagus in its own sauce and ate it hot with melted butter, salt, pepper and a dash of lemon. In various civilizations, it also played an important role as an aphrodisiac.

It is also used as an ornamental and medicinal plant. In Eastern Europe and Asia, decoctions of rhizomes and asparagus roots have long been used for the treatment of cardiovascular diseases, rheumatism and epilepsy. Traditionally, the roots were also used for non-specific inflammatory diseases of the kidney and bladder stones (irrigation therapy), liver diseases, dropsy, bronchial asthma and gout [3]. Compresses with decoctions were used to treat purulent skin pathologies. In Chinese folk medicine, decoctions of rhizomes and roots were used as diuretics, for improving mood, treating or preventing gout, diabetes, impotence, coughing with blood, dry mouth and throat, and constipation [3]. Asparagus has been recognized as an official medicinal plant in the pharmacopoeias of various countries, such as France, Mexico, Portugal and Venezuela. Although the species lost popularity in Europe during the Middle Ages, it later demonstrated a resurgence in the sixteenth century, when it became popular in England and France, and later in Germany.

Modern studies have shown it to have a diuretic effect and promote defecation; it also demonstrates high levels of basic nutrients, including vitamins, amino acids and mineral salts [4] and it is also rich in fiber [5,6]. Asparagus contains large amounts of folic acid (10 cooked shoots provide 225 micrograms, or almost 50% of the daily requirement) and vitamin C (10 cooked shoots provide 25 mg).

The present review describes the current literature concerning various biological properties (for example, antioxidant activity, antidiabetic activity, hypolipidemic activity, anticancer activity, and other properties) of various parts of *A. officinalis* L., especially its spears, and phytoconstituents isolated from asparagus. It is based on studies identified in electronic databases, including PubMed, ScienceDirect, Web of Knowledge, Sci Finder, Web of Science, and SCOPUS. The last search was run on 10 December 2023. The following terms were used: “*Asparagus*” or “*Asparagus officinalis*” or “asparagus shoots”, or “biological activity of asparagus” or “biological properties of asparagus”. Articles published before 2000 were excluded. The search was restricted to English language publications.

## 2. Morphology and Phytoconstituents of *A. officinalis*

Asparagus can grow to 100–150 cm tall with a stout stem. It has feathery needle-like leaves. Its flowers are arranged in clusters of four to fifteen flowers in a 6–32 nm long and 1 mm wide rosette. The root is indeterminate and fascicular. Its flowers are greenish white to yellowish. The fruits are small red berries (about 6–10 mm in diameter) and are poisonous for humans.

The plant is a source of various bioactive substances, which are mainly located in the lower portions of its spears and are discarded during industrial processing. Its main bioactive constituents are demonstrated in Figure 1. For example, the main saponins present in white and green *A. officinalis* are asparanin, protodioscin, yamogenin and sarsasapogenin. Al-Snafi et al. [3] report that *A. officinalis* also contains various steroid saponins, including asparagosides A, B, D, F, H and I, and the bitter steroid saponins.

Phenolic compounds have been the focal point of many analyses of asparagus. For example, Jimenez-Sanchez et al. [7] identified 32 phenolic compounds in green asparagus; one important flavonoid was rutin, representing 60–80% of the total phenolic compound content of purple and green asparagus extracts; for example, 1.51–7.29 mg/g dry weight for green asparagus, and below 0.5 mg/g dry weight for white asparagus [5,6,8,9,10]. Solana et al. [10] also note that rutin is the most abundant phenolic compound in whole *A. officinalis*.

Fan et al. [11] isolated the antioxidants isorhamnetin, quercetin and kaempferol from the residues of *A. officinalis*. They note that the extraction yield is significantly influenced by solvent composition, temperature and extraction time and propose that that optimal extraction occurs using 50% ethanol as a solvent with a liquid-solid ratio of 30:1, extraction temperature of 80 °C, and time of two hours. Phytochemical analysis using liquid chromatography/mass spectrometry (LC/MS)-MS by Comakli et al. [12] identified three phytochemicals in acetone extract from the aerial parts of *A. officinlis*: p-coumaric acid, rutin, and 4-hydroxybenzoic acid (284.3 ± 4.0; 135.4 ± 8.2, and 102.1 ± 5.5 µg analyte/g extract, respectively).

Ferulic acid, malic acid, isoferulic acid, citric acid, caffeic acid, asparagusic acid, and fumaric acid are typically extracted from dried *A. officinalis* roots [13].

One unique compound present in asparagus is asparagusic acid (1,2-dithiolane-4-carboxylic acid), a sulphur-containing compound [14,15] that appears to have a considerable influence on both the flavor and biological properties of asparagus. The distinguishing feature of asparagusic acid is its 1,2-dithiolane ring system [15,16]. Since then, Nakabayashi et al. [17] have detected other sulphur-containing compounds, including asparaptine in white and green asparagus spears.

The main phenolic compound in the methanolic extract of fresh *A. officinalis* roots was found to be to be caffeic acid accompanied by rutin, quercetin, various saponins, and four lignan types [18]. Other studies have found *A. officinalis* roots to be a valuable source of fructooligosaccharides [13].

Various carotenoid pigments were isolated from ripe and unripe *A. officinalis* stems, including lutein, zeaxanthin, capsanthin, capsorubin, violaxanthin, mutatoxanthin, neoxanthin and β-carotene [19].

Nutritional analyses have found *A. officinalis* to contain water (93.5%), carbohydrates (2.04%), proteins (1.91%), dietary fiber (1.31%), nitrogen (0.31%), and fat (0.16%) [3,20]. The nutritional values of raw and cooked asparagus are given in more detail in Table 2.

A recent study by Redondo-Cuence et al. [21] found *A. officinalis* and its by-products to have high nutritional and functional value. The findings not only indicate that the non-edible parts have high nutritional potential, but also that these parts may be used as prebiotics. In particular, the spear by-products can be used to promote the growth of commensal or probiotic lactobacillus and bifidobacteria strains. Moreover, the total dietary fiber values were found to be 58.1 ± 0.35 g/100 g dry matter for the spear by-products compared to only 23.8 ± 0.85 g/100 g dry matter for the edible portion. It is important to note that the recommended European consumption of fiber is estimated to be 20 g/person/day and supplementing the diet with spear by-products may increase fiber consumption. *A. officinalis* by-products may also be good sources of inulin and phenolic compounds. For example, in *A. officinalis* root by-products, the concentration of inulin was found to be 16.18 ± 0.24 g/100 g dry matter (1.3 ± 0.01 g/100 g dry matter for the spear edible portion), and caffeic acid 1.92 ± 0.06 mg/100 g dry matter; the latter compound was not identified in the spear edible portion.

It is important to note that various physical processing technologies can be used to extract bioactive substances in asparagus. For example, heating treatments can increase the antityrosine and antioxidant properties of asparagus; however, they also reduce the levels of heat-sensitive biologically active substances and can alter the green color. Moreover, edible coatings appear to significantly extend shelf life [22].

A recent study by Chitrakar et al. [23] described the protocols for extraction and purification of rutin from leafy by-products of *A. officinalis*. The crude extract was purified via multiple liquid–liquid back extraction using water, methanol or ethanol as a solvent. Fluorescence microscopy, Fourier transform infrared (FTIR) spectroscopy and LC-MS confirmed the purity of rutin to be 97.6%. More information about the phytochemical characteristics of *A. officinalis* have been described in other review papers [2,22,24].

Physical processing technologies, such as ultrasound-assisted extraction (UAE), supercritical fluid extraction (SFE) and pressurized liquid extraction (PLE), may also be effective at extracting bioactive compounds such as saponin, sugar and phenolic compounds.

## 3. Biological Properties of *A. officinalis* Extracts

### 3.1. Antioxidant Activity

Interestingly, in a study of 34 fruits and vegetables, *A. officinalis* was placed seventh as a radical scavenger and thirteenth in ferric-reducing power [25]. However, Vinson et al. [26] indicate that asparagus demonstrates the highest antioxidant properties among 23 commonly consumed vegetables based on dry weight.

The spear color of *A. officinalis* (green and white) reflects its phenolic compound content and thus its extract bioactivity, including 2,2-diphenyl-1-picrylhydrazyl (DPPH) radical scavenging [9]. Purple *A. officinalis* possesses stronger antioxidant properties than the white or green asparagus due to its abundant anthocyanins [11]. In addition, Khorasani et al. [27] indicate that in vivo grown *A. officinalis* extract demonstrates higher antioxidant capacity than in vitro grown plant extract.

In a study of the phenolic compounds of asparagus, Sun et al. [28,29] found that pectinase from *Aspergillus niger* reduces the antioxidant activity of its juice. In addition, Palfi et al. [30] report no significant differences in total phenolic compound content between wild and cultivated asparagus in Croatia; however, they found that the cultivated form demonstrated significantly higher antioxidant potential than wild asparagus.

Fathalipour et al. [31] indicate that hydroalcoholic root extract of *A. officinalis* (which is rich in flavonoids) has antioxidant properties. The extract demonstrated significantly higher IC_50_ (1117.65 ± 14.26 µg/mL) for DPPH scavenging activity than controls, and the extract (500 mg/kg) was found to have antinociceptive effects in formation and tail-flick tests in male Wistar rats.

Recently, the antioxidant activities of water, methanol, ethanol, chloroform, and acetone extracts of aerial parts of *A. officinalis* have been studied using various methods, including ABTS, DPPH, ferric ion reducing antioxidant power (FRAP), and Cu^2+^ ion reduction (CUPRAC). Comakli et al. [12] found the ethanol extract (10 µg/mL) to demonstrate the highest Fe^3+^ reduction capacity, and acetone extract (10 µg/mL) displayed the greatest Cu^2+^ reduction potential. Moreover, all tested extracts appeared to inhibit the activity of the enzymes acetylcholinesterase, butyrylcholinesterase, carbolic anhydrase and α-glucosidase, which are involved in various diseases, in vitro; it was also found that consumption of *A. officinalis* can support the treatment of certain diseases, including glaucoma, epilepsy, Alzheimer’s disease, and type 2 diabetes.

Recently, Alyami et al. [32] studied the protective action of *A. officinalis* against lead toxicity in mice (n = 26 mature male Swiss mice). Briefly, young shoots and leaves of *A. officinalis* were removed and air-dried in shadow at room temperature; following this, an aqueous extract was prepared. Unfortunately, however, the authors do not describe the chemical content of the plant extract. The study examined the effect of the aqueous extract on relieving testicular damage caused by lead acetate (PbAc) injection. Briefly, 200 mg/kg PbAc was injected into mice two hours after being supplemented with 400 mg/kg plant extract orally (for 14 days). The authors indicate that the *A. officinalis* extract reduced oxidative stress (testicular lipid peroxidation, nitric oxide concentration and glutathione content), thus protecting against apoptosis and inflammation. They also suggest that *A. officinalis* contains arginine, which is converted to nitric oxide: one of the most essential regulators of luteinizing hormone and follicle-stimulating hormone secretion.

### 3.2. Antidiabetic Activity

Zhao et al. [33] studied the hypoglycemic effect of the aqueous extract of *A. officinalis* by-products in a streptozotocin-induced diabetic rat model. Supplementation for 21 days significantly decreased serum glucose and triglyceride concentrations but increased hepatic glycogen concentration and body weight in diabetic rats. Similarly, Hafizur et al. [34,35] found *A. officinalis* seed extract to have anti-diabetic effects in non-obese type 2 diabetic rats supplemented with 250 and 500 mg/kg extract each day for 28 days; however, a particularly significant improvement was found for the 500 mg/kg dose, which was associated with an increase in insulin secretion. In addition, the authors noted that 0.5 mg/mL *A. officinalis* extract demonstrated 87% DPPH radical-scavenging activity in vitro, but only 32% inhibition of α-glucosidase in vitro. This result may suggest that the used extract has very little effect on delaying glucose absorption.

### 3.3. Hypolipidemic Activity

Two papers by Zhu et al. [36,37] found n-butanol extract from asparagus by-products to demonstrate hypolipidemic properties in mice fed a high-fat diet. For example, eight-week administration of the extract (40, 80, and 160 mg/kg body weight) decreased serum total cholesterol and low-density lipoprotein cholesterol and resulted in body weight gain. It also increased the levels of high-density lipoprotein and of various serum enzymes, including aspartate transaminase, alanine transaminase, alkaline phosphatase, and superoxide dismutase.

### 3.4. Anticancer Action

A review by Al-Snafi [3] found that *A. officinalis* may exert anticancer effects via various mechanisms, viz. by bestowing an antimutagenic effect, promoting cellular phase II detoxifying enzymes, inhibiting chronic inflammation, promoting healthier digestion and immune function, and finally, inhibiting oxidative stress. A recent study by Li et al. [38] examined the pharmacological mechanism of action of *A. officinalis* on multiple myeloma cells using bioinformatics tools (in vitro and in silico). The findings indicate that *A. officinalis* may exert anticancer activity by inhibiting the phosphoinositide 3-kinase (PI3K), protein kinase B (AKT) and nuclear factor kappa B (NF-κB) pathway. This would not only inhibit the proliferation and migration of myeloma cells, but also retard the tested cells in the G0/G1 phase and promote apoptosis.

### 3.5. Anti-Fungal and Antimicrobial Actions

Witaszek et al. [39] propose that ethanolic and methanolic fractions from white asparagus spears may have antifungal properties. This has been supported by other indicating potential anti-fungal [40,41] and antimicrobial effects [27,42].

### 3.6. Other Biological Activities

Other results suggest that the aqueous extract of *A. officinalis* roots can influence the hormone levels of the hypothalamic-pituitary-gonadal axis and the number of ovarian follicles in adult rats. For example, a 400 mg/kg dose increased progesterone and estrogen in female rats due to increased activity in the hypothalamic-pituitary axis [43].

Recently, Ho et al. [44] found that *A. officinalis* extract (0.5, 1, 5, and 10 mg/mL), commercially available as a supplement (ETAS^®^), has a synergic effect on progesterone synthesis with lipid droplets and mitochondrial function when combined with heat shock in bovine granulosa cells. The extract is believed to influence progesterone production in bovine granulosa cells by influencing heat shock protein 70 (HSP70), regulated activation of 3β-hydroxysteroid dehydrogenase and steroidogenic acute regulatory protein, and by improving mitochondrial activity and lipid metabolism.

Consumption of asparagus was found to alleviate various clinical symptoms of colitis, including stool blood, stool consistency, and spleen hypertrophy in C57BL/6 mice. The mice were supplemented with 2% cooked whole asparagus for three weeks [45].

## 4. Chemical Compounds Isolated from Asparagus and Their Properties

Steroidal saponins play an important function in both the biological and pharmacological properties of *A. officinalis*. *A. officinalis* saponins have been found to have hypolipidemic properties, decreasing low density lipoprotein (LDL) and total cholesterol concentrations in vivo [46,47]. In addition, saponins isolated from asparagus shoots have demonstrated anti-tumor and anti-fungal properties in vitro [8,18,28,48,49,50].

For example, Ji et al. [49] observed that saponins isolated from *A. officinalis* induce apoptosis in the HepG2 human hepatoma cell line through a mitochondrial-mediated pathway: they increased caspase-3 and -9 activity, down-regulated Bcl2 expression, up-regulated Bax expression, and induced cytochrome C release. After 72 h, apoptosis was found to be 30.9% (50 mg/L), 51.7% (100 mg/L) and 62.1% depending on the saponin concentration (200 mg/L).

Wang et al. [8] observed that saponins isolated from old stems of asparagus suppress tumor cell migration and invasion by targeting the Rho GTPase signaling pathway. At concentrations of 809.4 to 1830.0 µg/mL, the compounds inhibited the viability of breast, colon, and pancreatic cancer cells.

Shao et al. [51,52] also found crude saponins (75–100 µg/mL) isolated from the edible parts of *A. officinalis* to have anticancer properties. The compounds inhibited the growth of human leukemia HL-60 cells in vitro. The same authors observed that two oligofurostanosieds from the seeds of *A. officinalis* have cytotoxic effects [51,52]. Saponins also play an important role in the characteristic bitter taste of asparagus.

*A. officinalis* possesses various sulfur compounds, including asparagusis acid, which inhibit the activity of cyclooxygenase 2 (COX2): an enzyme associated with *inter alia* inflammation, carcinogenesis and cardiovascular diseases [53]. Nakabayashi et al. [17] found that another sulfur-containing compound isolated from white and green asparagus spears, asparaptine, inhibits angiotensin-converting enzyme (ACE) in vitro; this enzyme plays an important role in the regulation of hypertension in humans.

Huang et al. [18] also note that steroids isolated from the roots of *A. officinalis* appear to have cytotoxic effects.

Although *A. officinalis* extracts and their isolated components, especially saponins, appear to have various biological properties and pro-health potential, these observations are currently limited to in vitro and animal in vivo models. For example, various biological properties (including antioxidant, anti-cancer, antifungal, and antimicrobial properties) of *A. officinalis* spears were only observed in the in vitro model. For *A. officinalis* roots, anti-diabetic, antioxidant, hypolipidemic, and other activities were found in rat and mouse models (Table 3).

## 5. Conclusions

*A. officinalis* extracts and their isolated components, especially saponins, appear to have various biological properties and pro-health potential, including hypolipidemic activity, anti-tumor and antifungal property [55,56]. However, the efficacy, absorption and bioavailability of the bioactive compounds within *A. officinalis* have not been subjected to clinical studies. Furthermore, the interactions of the specific compounds isolated from *A. officinalis*, their biomolecular pathways and the mechanisms by which they influence health, especially in humans, remain very poorly defined, and their safety profile is unclear. As such, further studies are required.

## Figures and Tables

**Figure 1 foods-13-00288-f001:**
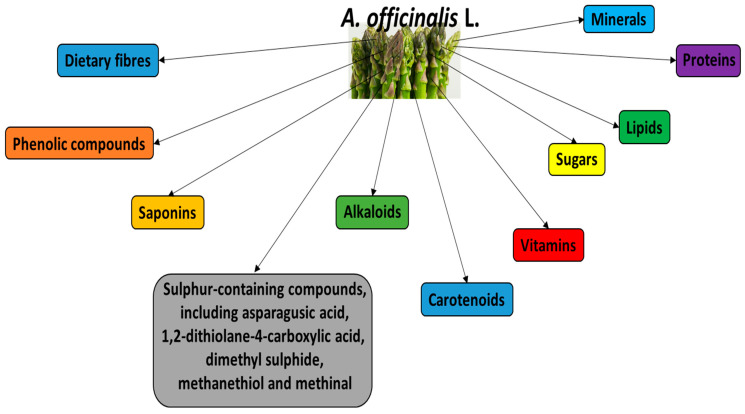
Main bioactive constituents of *A. officinalis* L.

**Table 1 foods-13-00288-t001:** The key odorants in cooked asparagus flavor ([2]; modified).

Chemical Compound	Aroma Attribute
dimethyl sulfide	Sulfurous, onion-like, fresh asparagus
2,3-butandione	Sweet, buttery, caramel
3-methylthio-propionanal	Sulfurous, cooked egg, cheesy, baked potato
2,3-pentanedione	Caramel, roasted nutty, buttery
trans-2-hexenal	Fruit, fresh asparagus
Hexanal	Woody, grass, fresh asparagus
2,6-dimethyl pyrazine	Woody, earthy, nutty, rusty, greasy
2-ethyl-3,5-dimethyl pyrazine	Coffee, nutty, roasted
2-methoxy-3-isopropyl pyrazine	Earthy
2,3-octanedione	Broccoli-like, buttery, dill-like
1-octen-3-ol	Mushroom-like, earthy
2-isobutyl-3-methoxypyrazine	Sprout-like, spicy, earthy
2-pentylfuran	Earthy, buttery

**Table 2 foods-13-00288-t002:** Nutritional value of raw and cooked asparagus (100 g) ([2]; modified).

Nutrients	Raw Asparagus	Cooked Asparagus
Calories	20 kcal	20 kcal
Dietary fiber	2.1 g	2.0 g
Sugars	1.9 g	1.3 g
Proteins	2.2 g	2.4 g
Lipids	0.12 g	0.22 g
Vitamin B_1_	0.143 mg	0.162 mg
Vitamin B_2_	0.141 mg	0.139 mg
Vitamin B_3_	0.978 mg	1.1 mg
Vitamin B_9_	52 µg	149 µg
Vitamin C	5.6 mg	7.7 mg
Vitamin E	1.13 mg	1.5 mg
Vitamin K	41.6 µg	50.6 µg
Calcium	24 mg	23 mg
Copper	0.19 mg	0.19 mg
Iron	2.14 mg	0.91 mg
Magnesium	14 mg	14 mg
Manganese	0.158 mg	0.158 mg
Potassium	202 mg	224 mg
Selenium	2.3 µg	10.8 µg
Sodium	2 mg	14 mg
Zinc	0.54 mg	0.54 mg

**Table 3 foods-13-00288-t003:** Biological properties of various parts of *A. officinalis* and its components in vivo and in vitro.

***A. officinalis* Part**	**Biological Property**	**References**
Roots	Anti-diabetic (in vivo—rat model)	[33]
	Antioxidant property (in vivo—rat model)	[31]
	Antinociceptive property (in vivo—rat model)	[31]
	Hypolipidemic property (in vivo—mouse model)	[36,37]
Spears	Antioxidant property (in vitro)	[12]
	Anti-cancer property (in vitro)	[38]
	Antifungal property (in vitro)	[27,39,42]
	Antimicrobial property (in vitro)	[27,42]
Shoots	Antioxidant property (in vivo mouse model)	[32]
Leaves	Antioxidant property (in vivo mouse model)	[32]
Seeds	Anti-diabetic (in vivo—rat model)	[34,35]
**Chemical Compounds Isolated from *A. officinalis***	**Biological Property**	**References**
Saponins	Hypolipidemic property (in vivo—rat model)	[46,47]
	Anti-tumor property (in vitro)	[8,18,28,29,48,49,50]
	Antifungal property (in vitro)	[8,18,28,29,48,49,50,54]
Oligofurostanosides	Cytotoxic property (in vitro)	[51]
Sulphur compounds	Inhibiting COX-2 (in vitro)	[53]
	Inhibiting ACE (in vitro)	[17]
Steroids	Cytotoxic property (in vitro)	[18]

## Data Availability

Not applicable.

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
