# Peer review of "A Review of the Pro-Health Activity of Asparagus officinalis L. and Its Components"

_foods, 2024, doi:10.3390/foods13020288_

Round 1

Reviewer 1 Report

Comments and Suggestions for Authors

Overall assessment: the article provides valuable information on the biological activities and compounds of Asparagus officinalis but could significantly enhance its quality by improving methodological transparency, critical analysis, discussing conflicting viewpoints, and providing more explicit search strategies. Some sections are more detailed than others, possibly leading to an imbalance in coverage. Ensuring uniform depth across different aspects might improve consistency. Improvements in these areas would greatly strengthen the article's credibility and usefulness as a comprehensive review.

Introduction:

·         Background: the introduction provides a brief overview of the biological activities of different parts of asparagus officinalis, its compounds, and their potential health benefits. It touches upon its significance in health and medicine.

·         Objectives: the objectives are implied but not explicitly stated. The article aims to present the biological properties and potential health benefits of asparagus components.

Literature search and methodology:

·         Search strategy: the article lacks explicit details on the literature search strategy, including databases, keywords, or inclusion/exclusion criteria used.

·         Selection criteria: information regarding the process used to select articles is absent, affecting the transparency and reliability of the review.

Organization and structure:

·         Logical flow: the article follows a logical sequence, organizing content into sections based on different biological properties and compounds.

·         Thematic segmentation: it organizes information around themes like biological activities, specific compounds, and their properties.

Comprehensive coverage:

·         Inclusion of key studies: it includes key studies relevant to asparagus's biological properties and compounds, citing various sources.

·         Critical analysis: the review lacks a critical analysis, such as evaluating the strengths, weaknesses, and contributions of the included studies.

·         Coverage of diverse perspectives: It covers diverse biological properties and compounds but might benefit from discussing conflicting findings or varying viewpoints within the field.

Synthesis and discussion:

·         Synthesize findings: The article attempts to bring together the diverse biological activities and compound properties of Asparagus officinalis. However, it could strengthen this synthesis by highlighting overarching patterns, trends, or inconsistencies across studies. For instance, are there recurring themes in the effects of different compounds? Are there conflicting findings that need reconciliation?

·         Discussion of implications: While the article touches on potential implications for future research, it does not thoroughly explore the identified gaps or propose specific areas for further exploration. It could benefit from a more detailed exploration of the limitations in the current research and propose precise directions for future studies. For example, are there particular compounds whose mechanisms of action need more investigation? Are there specific health conditions where the application of asparagus compounds could be promising but needs more study?

Conclusion:

·         The conclusion encapsulates the biological properties and compounds of Asparagus officinalis and acknowledges the necessity for further research. However, it could be more detailed by summarizing the major findings and their implications on health or medicine.

·         While the conclusion provides some insights, it lacks detailed recommendations based on the review's findings. Offering concrete suggestions for future studies, potential applications in medicine, or areas for experimentation could strengthen the conclusion.

Visual aids: Although the article does include one figure and three tables, their presence falls short in maximizing the visual representation that could enhance comprehension. The available visual aids could be more strategically employed to illustrate complex concepts, data correlations, or connections between variables. By optimizing the use of these existing visual elements or potentially introducing additional, more illustrative figures, graphs, or diagrams, the article could significantly elevate its visual appeal and effectively communicate intricate information. Strengthening the visual aids in this manner would substantially augment the article's comprehensiveness, engagement, and overall informativeness for a broader readership.

Comments on the Quality of English Language

The English language quality in the article, it generally maintains a good standard. The language is predominantly clear, utilizing scientific terminology appropriately. The sentences are well-structured, facilitating comprehension for both experts and non-experts. However, there are occasional instances where sentence structures could be improved for smoother readability, and minor grammatical issues might be present, although they don't significantly impact overall understanding. It's important to ensure consistency in style and tone throughout the article for a more polished presentation. Overall, the English language quality is good but could benefit from some refinement for greater coherence and fluidity.

Author Response

Overall assessment: the article provides valuable information on the biological activities and compounds of Asparagus officinalis but could significantly enhance its quality by improving methodological transparency, critical analysis, discussing conflicting viewpoints, and providing more explicit search strategies. Some sections are more detailed than others, possibly leading to an imbalance in coverage. Ensuring uniform depth across different aspects might improve consistency. Improvements in these areas would greatly strengthen the article's credibility and usefulness as a comprehensive review.

Thank you for your helpful comments. All of them have been taken into consideration when revising the manuscript.

Introduction:

  • Background: the introduction provides a brief overview of the biological activities of different parts of asparagus officinalis, its compounds, and their potential health benefits. It touches upon its significance in health and medicine.
  • Objectives: the objectives are implied but not explicitly stated. The article aims to present the biological properties and potential health benefits of asparagus components.

Response: I have corrected the article aims. Now, it is “The present review describes the current literature concerning the pro-health properties of various parts of A. officinalis L., especially its spears, and phytoconstituents isolated from asparagus.”

Literature search and methodology:

  • Search strategy: the article lacks explicit details on the literature search strategy, including databases, keywords, or inclusion/exclusion criteria used.

Response: I have corrected this information. Now, it is: “The present review describes the current literature concerning the pro-health properties of various parts of A. officinalis L., especially its spears, and phytoconstituents isolated from asparagus. It is based on studies identified in electronic databases, including PubMed, ScienceDirect, Web of Knowledge, Sci Finder, Web of Science, and SCOPUS. The last search was run on 10th December 2023. The following terms were used: “Asparagus” or“Asparagus officinalis” orasparagus organs”, or “biological activity of asparagus”. The search was restricted to English language publications.”

  • Selection criteria: information regarding the process used to select articles is absent, affecting the transparency and reliability of the review.

Response: I have corrected this information. Now, it is: “The present review describes the current literature concerning the pro-health properties of various parts of A. officinalis L., especially its spears, and phytoconstituents isolated from asparagus. It is based on studies identified in electronic databases, including PubMed, ScienceDirect, Web of Knowledge, Sci Finder, Web of Science, and SCOPUS. The last search was run on 10th December 2023. The following terms were used: “Asparagus” or“Asparagus officinalis” orasparagus organs”, or “biological activity of asparagus”. The search was restricted to English language publications.

  •  

Organization and structure:

  • Logical flow: the article follows a logical sequence, organizing content into sections based on different biological properties and compounds.
  • Thematic segmentation: it organizes information around themes like biological activities, specific compounds, and their properties.

Comprehensive coverage:

  • Inclusion of key studies: it includes key studies relevant to asparagus's biological properties and compounds, citing various sources.
  • Critical analysis: the review lacks a critical analysis, such as evaluating the strengths, weaknesses, and contributions of the included studies.

Response: I have added more information about it in 4 and 5 sections, especially in the chapter of Conclusion. For example: “Although A. officinalis extracts and their isolated components, especially saponins, appear to have various biological properties and pro-health potential, these observations are currently limited to in vitro and animal in vivo models (Table 3). In addition, the efficacy, absorption and bioavailability of the bioactive compounds within A. officinalis have not been subjected to clinical studies. Furthermore, the interactions of the specific compounds isolated from A. officinalis, their biomolecular pathways and the mechanisms by which they influence health, especially in humans, remain very poorly defined, and their safety profile is unclear. As such, further studies are required. Before A. officinalis extracts can be used as phytosupplements, they need to be standardized in accordance with the standard protocol for concentration, which has pro-health properties.”

  • Coverage of diverse perspectives: It covers diverse biological properties and compounds but might benefit from discussing conflicting findings or varying viewpoints within the field.

Response: I have added more information about it in 4 and 5 sections, especially Conclusion. For example: “Although A. officinalis extracts and their isolated components, especially saponins, appear to have various biological properties and pro-health potential, these observations are currently limited to in vitro and animal in vivo models (Table 3). In addition, the efficacy, absorption and bioavailability of the bioactive compounds within A. officinalis have not been subjected to clinical studies. Furthermore, the interactions of the specific compounds isolated from A. officinalis, their biomolecular pathways and the mechanisms by which they influence health, especially in humans, remain very poorly defined, and their safety profile is unclear. As such, further studies are required. Before A. officinalis extracts can be used as phytosupplements, they need to be standardized in accordance with the standard protocol for concentration, which has pro-health properties.”

  •  

Synthesis and discussion:

  • Synthesize findings: The article attempts to bring together the diverse biological activities and compound properties of Asparagus officinalis. However, it could strengthen this synthesis by highlighting overarching patterns, trends, or inconsistencies across studies. For instance, are there recurring themes in the effects of different compounds? Are there conflicting findings that need reconciliation?

Response: Various biological properties of A. officinalis extracts were described in 3 section. Moreover, chemical compounds isolated from asparagus and their properties were described in 4 section. In addition, Table 3 describes biological properties of various parts of A. officinalis and its components in vivo and in vitro. In the end of 4 section and conclusion, I have also demonstrated: “Although A. officinalis extracts and their isolated components, especially saponins, appear to have various biological properties and pro-health potential, these observations are currently limited to in vitro and animal in vivo models (Table 3). In addition, the efficacy, absorption and bioavailability of the bioactive compounds within A. officinalis have not been subjected to clinical studies. Furthermore, the interactions of the specific compounds isolated from A. officinalis, their biomolecular pathways and the mechanisms by which they influence health, especially in humans, remain very poorly defined, and their safety profile is unclear. As such, further studies are required. Before A. officinalis extracts can be used as phytosupplements, they need to be standardized in accordance with the standard protocol for concentration, which has pro-health properties.”

  • Discussion of implications: While the article touches on potential implications for future research, it does not thoroughly explore the identified gaps or propose specific areas for further exploration. It could benefit from a more detailed exploration of the limitations in the current research and propose precise directions for future studies. For example, are there particular compounds whose mechanisms of action need more investigation? Are there specific health conditions where the application of asparagus compounds could be promising but needs more study?

Response: I have added more information about it in 4 and 5 sections, especially Conclusion. For example: “Although A. officinalis extracts and their isolated components, especially saponins, appear to have various biological properties and pro-health potential, these observations are currently limited to in vitro and animal in vivo models (Table 3). In addition, the efficacy, absorption and bioavailability of the bioactive compounds within A. officinalis have not been subjected to clinical studies. Furthermore, the interactions of the specific compounds isolated from A. officinalis, their biomolecular pathways and the mechanisms by which they influence health, especially in humans, remain very poorly defined, and their safety profile is unclear. As such, further studies are required. Before A. officinalis extracts can be used as phytosupplements, they need to be standardized in accordance with the standard protocol for concentration, which has pro-health properties.”

  • Conclusion:
  • The conclusion encapsulates the biological properties and compounds of Asparagus officinalis and acknowledges the necessity for further research. However, it could be more detailed by summarizing the major findings and their implications on health or medicine.
  • While the conclusion provides some insights, it lacks detailed recommendations based on the review's findings. Offering concrete suggestions for future studies, potential applications in medicine, or areas for experimentation could strengthen the conclusion.

Response: I have modified the chapter of Conclusion. Now, it is: “A. officinalis extracts and their isolated components, especially saponins, appear to have various biological properties and pro-health potential, including hypolipidemic activity, anti-tumor and antifungal property. However, the efficacy, absorption and bioavailability of the bioactive compounds within A. officinalis have not been subjected to clinical studies. Furthermore, the interactions of the specific compounds isolated from A. officinalis, their biomolecular pathways and the mechanisms by which they influence health, especially in humans, remain very poorly defined, and their safety profile is unclear. As such, further studies are required. Before A. officinalis extracts can be used as phytosupplements, they need to be standardized in accordance with the standard protocol for concentration, which has pro-health properties.”

Visual aids: Although the article does include one figure and three tables, their presence falls short in maximizing the visual representation that could enhance comprehension. The available visual aids could be more strategically employed to illustrate complex concepts, data correlations, or connections between variables. By optimizing the use of these existing visual elements or potentially introducing additional, more illustrative figures, graphs, or diagrams, the article could significantly elevate its visual appeal and effectively communicate intricate information. Strengthening the visual aids in this manner would substantially augment the article's comprehensiveness, engagement, and overall informativeness for a broader readership.

Response: I have not decided to prepare new figure, because the interactions of the specific compounds isolated from A. officinalis, their biomolecular pathways and the mechanisms by which they influence health, especially in humans, remain very poorly defined. Table 3 describes biological properties of various parts of A. officinalis and its components in vivo and in vitro.

Comments on the Quality of English Language

The English language quality in the article, it generally maintains a good standard. The language is predominantly clear, utilizing scientific terminology appropriately. The sentences are well-structured, facilitating comprehension for both experts and non-experts. However, there are occasional instances where sentence structures could be improved for smoother readability, and minor grammatical issues might be present, although they don't significantly impact overall understanding. It's important to ensure consistency in style and tone throughout the article for a more polished presentation. Overall, the English language quality is good but could benefit from some refinement for greater coherence and fluidity.

Response: English was corrected by native speaker (Adhoc English Edward Lowczowski).

Reviewer 2 Report

Comments and Suggestions for Authors

The manuscript titled “A review of the pro-health activity of Asparagus officinalis L. and its components” written by Beata Olas discussed the various therapeutic potential of  Asparagus officinalis  along with its bioactive compounds and functions.  However, I feel the manuscript has several flows and as per the standard of the Journal’s as well as Foods readers, this manuscript is not suitable for publication.  

Various literatures are available detailing the benefits of Asparagus officinalis and author presents very basic information. I feel author do not bring any new information to this manuscript. Additionally, the presentation of the manuscript is very poor. There are many sentences, which do not have any references.

Comments on the Quality of English Language

Minor 

Author Response

The manuscript titled “A review of the pro-health activity of Asparagus officinalis L. and its components” written by Beata Olas discussed the various therapeutic potential of  Asparagus officinalis  along with its bioactive compounds and functions.  However, I feel the manuscript has several flows and as per the standard of the Journal’s as well as Foods readers, this manuscript is not suitable for publication. 

Various literatures are available detailing the benefits of Asparagus officinalis and author presents very basic information. I feel author do not bring any new information to this manuscript. Additionally, the presentation of the manuscript is very poor. There are many sentences, which do not have any references.

Response: Thank you for your helpful comments. All of them have been taken into consideration when revising the manuscript. For example,

I have corrected the article aims. Now, it is “The present review describes the current literature concerning the pro-health properties of various parts of A. officinalis L., especially its spears, and phytoconstituents isolated from asparagus.”

I have added more information about literature search and methodology: Now, it is: “The present review describes the current literature concerning the pro-health properties of various parts of A. officinalis L., especially its spears, and phytoconstituents isolated from asparagus. It is based on studies identified in electronic databases, including PubMed, ScienceDirect, Web of Knowledge, Sci Finder, Web of Science, and SCOPUS. The last search was run on 10th December 2023. The following terms were used: “Asparagus” or“Asparagus officinalis” orasparagus organs”, or “biological activity of asparagus”. The search was restricted to English language publications.”

In addition, I have modified the chapter of Conclusion. Now, it is: “A. officinalis extracts and their isolated components, especially saponins, appear to have various biological properties and pro-health potential, including hypolipidemic activity, anti-tumor and antifungal property. However, the efficacy, absorption and bioavailability of the bioactive compounds within A. officinalis have not been subjected to clinical studies. Furthermore, the interactions of the specific compounds isolated from A. officinalis, their biomolecular pathways and the mechanisms by which they influence health, especially in humans, remain very poorly defined, and their safety profile is unclear. As such, further studies are required. Before A. officinalis extracts can be used as phytosupplements, they need to be standardized in accordance with the standard protocol for concentration, which has pro-health properties.”

  •  

Reviewer 3 Report

Comments and Suggestions for Authors

The manuscript examines the pro-health properties and chemical composition of Asparagus officinalis. Having explored the information on this type of food-related product could contribute to its utilization and promotion. In spite of this, the following suggestions will probably catch up with the advancements in the field.

 (1) The abstract lacks entirety and logic. It is necessary to describe the significance of this work.

 (2) In the INTRODUCTION, what are the opportunities and challenges associated with Asparagus officinalis? The sentences (Lines 83-85) should also be deleted.

 (3) Line 86, for morphology, the content is insufficient.

 (4) In Figure 1, what is the rationale for the sequence of bioactive constituents?

 (5) In line 169, a number of biological activities have been introduced; however, their modes of action remain to be explored.

 (6) Line 266, it is suggested that they should be made dependent in order to form a paragraph.

 (7) Line 296, the Table 3 should be removed from the conclusion and added to Section 3.

 (8) Remove Lines 321-338.

 (9) Update the tables and figures quality.

 (10) Many bugs in the references.

Author Response

The manuscript examines the pro-health properties and chemical composition of Asparagus officinalis. Having explored the information on this type of food-related product could contribute to its utilization and promotion. In spite of this, the following suggestions will probably catch up with the advancements in the field.

Thank you for your helpful comments. All of them have been taken into consideration when revising the manuscript.

  • The abstract lacks entirety and logic. It is necessary to describe the significance of this work.

Response: I have corrected Abstract. Now, it is: “The genus Asparagus comprises about 300 species, including A. curilus, A. filicinus, A. reacemosus, and A. officinalis L.. A particularly well-known member of the genus is Asparagus officinalis L., also known as “the king of vegetables”. Consuming A. officinalis makes an excellent contribution to a healthy diet. Modern studies have shown it to have a diuretic effect and promote defecation; it also demonstrates high levels of basic nutrients, including vitamins, amino acids and mineral salts. Moreover, it is rich in fibre. Asparagus contains large amounts of folic acid (10 cooked shoots provide 225 micrograms, or almost 50% of the daily requirement) and vitamin C (10 cooked shoots provide 25 mg). The present review describes the current literature concerning the pro-health properties of various parts of A. officinalis L., with a particular focus on its spears. It is based on studies identified in electronic databases, including PubMed, ScienceDirect, Web of Knowledge, Sci Finder, Web of Science, and SCOPUS. The data indicates that the various parts of A. officinalis, especially the spears, contain many bioactive compounds. However, although the extracts and chemical compounds isolated from A. officinalis, especially saponins, appear to have various biological properties and pro-health potential, these observations are limited to in vitro and animal in vivo models.”

  • In the INTRODUCTION, what are the opportunities and challenges associated with Asparagus officinalis? The sentences (Lines 83-85) should also be deleted.

Response: I have corrected the chapter of Introduction, for example, I have modified the sentences (lines 83-85) and I have added more information. Now, it is: “The present review describes the current literature concerning the pro-health properties of various parts of A. officinalis L., especially its spears, and phytoconstituents isolated from asparagus. It is based on studies identified in electronic databases, including PubMed, ScienceDirect, Web of Knowledge, Sci Finder, Web of Science, and SCOPUS. The last search was run on 10th December 2023. The following terms were used: “Asparagus” or“Asparagus officinalis” orasparagus organs”, or “biological activity of asparagus”. The search was restricted to English language publications.”

  • Line 86, for morphology, the content is insufficient.

Response: I have added more information about it. Now, it is: “Asparagus can grow to 100-150 cm tall with a stout stem. It has feathery needle-like leaves. Its flowers are arranged in clusters of four to 15 flowers in a 6-32 nm-long and 1 mm-wide rosette. The root is indetermine and fascicular. Its flowers are greenish-white to yellowish. The fruits are small red berries (about 6 – 10 mm in diameter) and poisonous for humans.”

  • In Figure 1, what is the rationale for the sequence of bioactive constituents?

Response: A. officinalis is a source of various bioactive substances, which are mainly located in the lower portions of its spears and are discarded during the industrial processing, and its main bioactive constituents are demonstrated in Figure 1. More information about these phytoconstituents are described in the manuscript and in Table 2.

  • In line 169, a number of biological activities have been introduced; however, their modes of action remain to be explored.

Response: The chapter “Biological properties….” describes various biological properties of A. officinalis in various models (in vitro and in vivo). On the other hand, chapter of Introduction describes only short information about the pro-healthy activity of this plant.

  • Line 266, it is suggested that they should be made dependent in order to form a paragraph.

Response: I have corrected. Now, it is new chapter: “Chemical compounds isolated from asparagus and their properties”.

  • Line 296, the Table 3 should be removed from the conclusion and added to Section 3.

Response: I have added the Table 3 in Section 4 - “Chemical compounds isolated from asparagus and their properties”.

  • Remove Lines 321-338.

Response: I have not removed these lines, because the Editor added these.

  • Update the tables and figures quality.

Response: I have corrected them.

 (10) Many bugs in the references.

Response: I have corrected them.

Round 2

Reviewer 1 Report

Comments and Suggestions for Authors

Dear Authors,

Thank you for the revised manuscript. In this round of revisions, several improvements have been made, but there are still some crucial aspects that require further attention.

The introduction remains somewhat unstructured, lacking a clear statement of objectives. It briefly discusses the biological activities of Asparagus officinalis but does not distinctly outline the research aims. Additionally, the methodology section requires more explicit details regarding the search strategy and inclusion/exclusion criteria to enhance transparency and reliability.

While the manuscript covers various biological properties and compounds of asparagus from diverse sources, it lacks critical analysis. It would greatly benefit from evaluating the strengths, weaknesses, and contributions of the included studies. Addressing conflicting findings or differing viewpoints within the field would significantly enrich the content, providing a more comprehensive overview.

Regarding the discussion, although the diverse bioactivities and compounds of Asparagus officinalis are acknowledged, there's room for improvement. Emphasizing recurring patterns or inconsistencies across studies could elevate the synthesis. Furthermore, exploring identified gaps in greater depth and proposing specific areas for future research would strengthen the manuscript's contribution to the field.

The revised conclusion has addressed some previously highlighted shortcomings, discussing the biological properties and compounds of A. officinalis and acknowledging the need for further research. However, it lacks a detailed synthesis of major findings and their implications for health or medicine. Providing specific recommendations based on the reviewed findings and suggesting potential experimental directions or medical applications would enhance the conclusion.

Overall, the manuscript has progressed, but addressing these remaining points will significantly enhance its quality and contribution to the field.

Comments on the Quality of English Language

The English language in the paper is generally proficient, but there are areas where clarity and precision could be improved:

·         Sentence Structure: Some sentences are overly long and complex, which can make it challenging to follow the intended meaning. Breaking them into smaller, more digestible sentences could enhance readability.

(Original: "The experiment, which aimed to evaluate the correlation between variables X and Y through a series of controlled conditions while also considering the impact of variable Z, demonstrated intriguing results." Suggested Improvement: "The experiment aimed to evaluate the correlation between variables X and Y in controlled conditions. Additionally, it considered the impact of variable Z, revealing intriguing results.")

·         Punctuation and Grammar: There are instances where the use of commas might be excessive, leading to slightly convoluted sentences. A review for comma placement could help improve the flow and clarity.

(Original: "The data, collected over a span of three years, which showed consistent patterns, indicates a strong relationship between the two factors." Suggested Improvement: "The data, collected over three years and demonstrating consistent patterns, indicate a strong relationship between the two factors.")

·         Word Choice: There are a few instances where word choices might be more precise or where technical terms could be better explained for a broader audience.

Author Response

Reviewer 1

Thank you for the revised manuscript. In this round of revisions, several improvements have been made, but there are still some crucial aspects that require further attention.

Thank you for your helpful comments. All of them have been taken into consideration when revising the manuscript.

The introduction remains somewhat unstructured, lacking a clear statement of objectives. It briefly discusses the biological activities of Asparagus officinalis but does not distinctly outline the research aims. Additionally, the methodology section requires more explicit details regarding the search strategy and inclusion/exclusion criteria to enhance transparency and reliability.

Response: I have corrected this information. Now, it is: “The present review describes the current literature concerning various biological properties (for example antioxidant activity, antidiabetic activity, hypolipidemic activity, anticancer activity, and other) of various parts of A. officinalis L., especially its spears, and phytoconstituents isolated from asparagus. It is based on studies identified in electronic databases, including PubMed, ScienceDirect, Web of Knowledge, Sci Finder, Web of Science, and SCOPUS. The last search was run on 10th December 2023. The following terms were used: “Asparagus” or “Asparagus officinalis” orasparagus shoots”, or “biological activity of asparagus” or “biological properties of asparagus”. Articles published before 2000 were excluded. The search was restricted to English language publications.

While the manuscript covers various biological properties and compounds of asparagus from diverse sources, it lacks critical analysis. It would greatly benefit from evaluating the strengths, weaknesses, and contributions of the included studies. Addressing conflicting findings or differing viewpoints within the field would significantly enrich the content, providing a more comprehensive overview.

Response: I have added short information about it in the end of 4th chapter: “Although A. officinalis extracts and their isolated components, especially saponins, appear to have various biological properties and pro-health potential, these observations are currently limited to in vitro and animal in vivo models. For example, various biological properties (including antioxidant, anti-cancer, antifungal, and antimicrobial properties) of A. officinalis spears were only observed in in vitro model. For A. officinalis roots, anti-diabetic, antioxidant, hypolipidemic, and other activities were found in rat and mouse models (Table 3).”

Moreover, I have added this information in the chapter of Conclusion: “A. officinalis extracts and their isolated components, especially saponins, appear to have various biological properties and pro-health potential, including hypolipidemic activity, anti-tumor and antifungal property. However, the efficacy, absorption and bioavailability of the bioactive compounds within A. officinalis have not been subjected to clinical studies. Furthermore, the interactions of the specific compounds isolated from A. officinalis, their biomolecular pathways and the mechanisms by which they influence health, especially in humans, remain very poorly defined, and their safety profile is unclear. As such, further studies are required. Before A. officinalis extracts can be used as phytosupplements in the prophylaxis and treatment of various diseases, including cardiovascular diseases, diabetic, cancer, ant other, they need to be standardized in accordance with the standard protocol for concentration, which has pro-health properties.”

Regarding the discussion, although the diverse bioactivities and compounds of Asparagus officinalis are acknowledged, there's room for improvement. Emphasizing recurring patterns or inconsistencies across studies could elevate the synthesis. Furthermore, exploring identified gaps in greater depth and proposing specific areas for future research would strengthen the manuscript's contribution to the field.

Response: I have added short information about it in the end of 4th chapter: “Although A. officinalis extracts and their isolated components, especially saponins, appear to have various biological properties and pro-health potential, these observations are currently limited to in vitro and animal in vivo models. For example, various biological properties (including antioxidant, anti-cancer, antifungal, and antimicrobial properties) of A. officinalis spears were only observed in in vitro model. For A. officinalis roots, anti-diabetic, antioxidant, hypolipidemic, and other activities were found in rat and mouse models (Table 3).”

In addition, I have added more information about it in the chapter of Conclusion: “A. officinalis extracts and their isolated components, especially saponins, appear to have various biological properties and pro-health potential, including hypolipidemic activity, anti-tumor and antifungal property. However, the efficacy, absorption and bioavailability of the bioactive compounds within A. officinalis have not been subjected to clinical studies. Furthermore, the interactions of the specific compounds isolated from A. officinalis, their biomolecular pathways and the mechanisms by which they influence health, especially in humans, remain very poorly defined, and their safety profile is unclear. As such, further studies are required. Before A. officinalis extracts can be used as phytosupplements in the prophylaxis and treatment of various diseases, including cardiovascular diseases, diabetic, cancer, ant other, they need to be standardized in accordance with the standard protocol for concentration, which has pro-health properties.”

The revised conclusion has addressed some previously highlighted shortcomings, discussing the biological properties and compounds of A. officinalis and acknowledging the need for further research. However, it lacks a detailed synthesis of major findings and their implications for health or medicine. Providing specific recommendations based on the reviewed findings and suggesting potential experimental directions or medical applications would enhance the conclusion.

Response: I have added more information about it in the chapter of Conclusion: “A. officinalis extracts and their isolated components, especially saponins, appear to have various biological properties and pro-health potential, including hypolipidemic activity, anti-tumor and antifungal property. However, the efficacy, absorption and bioavailability of the bioactive compounds within A. officinalis have not been subjected to clinical studies. Furthermore, the interactions of the specific compounds isolated from A. officinalis, their biomolecular pathways and the mechanisms by which they influence health, especially in humans, remain very poorly defined, and their safety profile is unclear. As such, further studies are required. Before A. officinalis extracts can be used as phytosupplements in the prophylaxis and treatment of various diseases, including cardiovascular diseases, diabetic, cancer, ant other, they need to be standardized in accordance with the standard protocol for concentration, which has pro-health properties.”

Overall, the manuscript has progressed, but addressing these remaining points will significantly enhance its quality and contribution to the field.

Comments on the Quality of English Language

The English language in the paper is generally proficient, but there are areas where clarity and precision could be improved:

  • Sentence Structure: Some sentences are overly long and complex, which can make it challenging to follow the intended meaning. Breaking them into smaller, more digestible sentences could enhance readability.

(Original: "The experiment, which aimed to evaluate the correlation between variables X and Y through a series of controlled conditions while also considering the impact of variable Z, demonstrated intriguing results." Suggested Improvement: "The experiment aimed to evaluate the correlation between variables X and Y in controlled conditions. Additionally, it considered the impact of variable Z, revealing intriguing results.")

  • Punctuation and Grammar: There are instances where the use of commas might be excessive, leading to slightly convoluted sentences. A review for comma placement could help improve the flow and clarity.

(Original: "The data, collected over a span of three years, which showed consistent patterns, indicates a strong relationship between the two factors." Suggested Improvement: "The data, collected over three years and demonstrating consistent patterns, indicate a strong relationship between the two factors.")

  • Word Choice: There are a few instances where word choices might be more precise or where technical terms could be better explained for a broader audience.

Response: English in the manuscript was corrected by Adhoc English Edward Lowczowski, E-mail: ed.lowczowski@adhocenglish.pl WWW: adhocenglish.pl.

Reviewer 3 Report

Comments and Suggestions for Authors

If possible, please update the Figure 1. 

Author Response

If possible, please update the Figure 1. 

Thank you for your helpful comments. However, I have not modified Fig. 1 - Main bioactive constituents of A. officinalis L., because Table 2 describes more information about nutritional value of raw and cooked asparagus.